# WASP: A Weight-Space Approach to Detecting Learned Spuriousness

Cristian Daniel Păduraru[1,2]     Antonio Bărbălau[1,2]     Radu Filipescu[1,2]

Andrei Liviu Nicolicioiu[3,4]     Elena Burceanu[1]

[1]Bitdefender, Romania
[2]University of Bucharest, Romania
[3]Mila, Montreal, Canada
[4]University of Montreal, Canada
{cpaduraru, ext-abarbalau, eburceanu}@bitdefender.com
andrei.nicolicioiu@mila.quebec

## Abstract

It is of crucial importance to train machine learning models such that they clearly understand what defines each class in a given task. Though there is a sum of works dedicated to identifying the spurious correlations featured by a dataset that may impact the model's understanding of the classes, all current approaches rely solely on data or error analysis. That is, they cannot point out spurious correlations learned by the model that are not already pointed out by the counterexamples featured in the validation or training sets. We propose a method that transcends this limitation, switching the focus from analyzing a model's predictions to analyzing the model's weights, the mechanism behind the making of the decisions, which proves to be more insightful. Our proposed **W**eight-space **A**pproach to detecting **Sp**uriousness (**WASP**) relies on analyzing the weights of foundation models as they drift towards capturing various (spurious) correlations while being fine-tuned on a given dataset. We demonstrate that different from previous works, our method (i) can expose spurious correlations featured by a dataset even when they are not exposed by training or validation counterexamples, (ii) it works for multiple modalities such as image and text, and (iii) it can uncover previously untapped spurious correlations learned by ImageNet-1k classifiers.

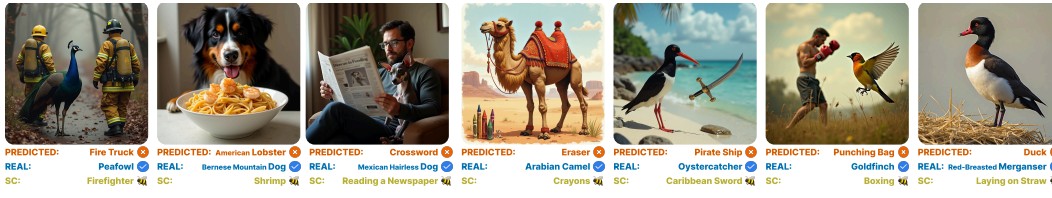

Figure 1: Qualitative results depicting a series of Spurious Correlations (SCs), uncovered employing our proposed WASP approach for CLIP ViT-L/14 [35] fine-tuned on ImageNet-1k [8]. In each scenario, a (real) ImageNet class is depicted alongside a concept found to be positively correlated with a different (predicted) class. Though in every scenarios a single ImageNet-1k class is clearly depicted, the model predicts a class that is not illustrated at all, clinging to the learned SCs.

39th Conference on Neural Information Processing Systems (NeurIPS 2025) Workshop: Reliable ML from Unreliable Data.

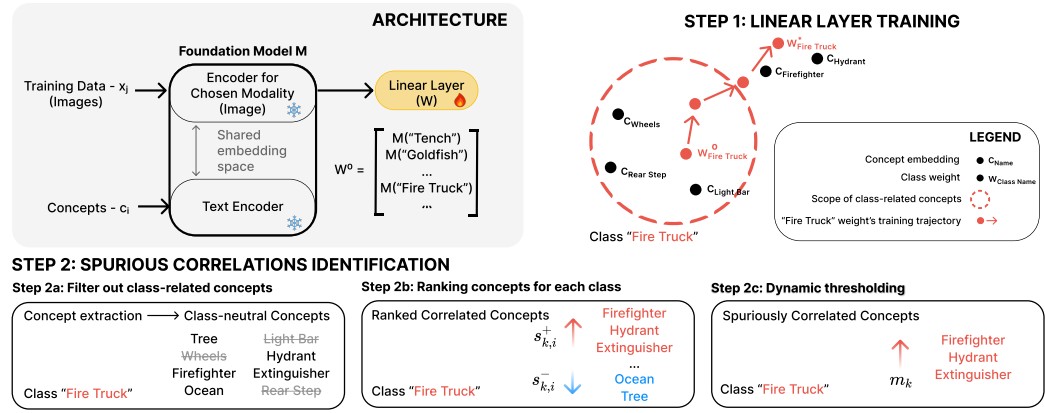

Figure 2: Following WASP's steps for the ImageNet-1k "Fire Truck" class. In Step 1, during training, the classification weights $W$ drift from the initial class concept embedding $W^0$, outside the scope of relevant concepts, towards spuriously correlated ones. In Step 2, our method filters out class-related concepts and, using an embedding-space scoring system, ranks and automatically marks the highest-ranking class-neutral concepts as SCs.

# 1  Introduction

Deep neural networks, and especially fine-tuned versions of foundation models, are commonly deployed in critical areas such as healthcare, finance, and criminal justice, where decisions based on **spurious correlations (SCs)** can have significant societal consequences [1, 6]. Even if the pretrained model has been validated by the community, the dataset leveraged in the fine-tuning process can, and usually does, imprint the model with new SCs. We present such examples in Fig. 1, where a CLIP model, fine-tuned on ImageNet-1k, labels the first image as a "Fire Truck", even in the complete absence of any fire truck or truck-related features and even given the presence of a peafowl (which is an ImageNet-1k class) right in front and center. This is due to the fact that, as our proposed WASP protocol discovered, the model spuriously correlates firefighters (which are not an ImageNet-1K class) with the "Fire Truck" class.

Subpopulation shift setups [5, 26, 50], featuring naturally occurring SCs, provide researchers with a controlled environment for studying SC detection and prevention. Within this context, recent efforts have begun to employ foundation models [4, 17, 58] in their investigation protocols. Some of these efforts [4, 58] focused on finding SCs through data analysis, without referencing model predictions. While these findings are valuable, they may not always be relevant. For example, an SC may not impact a machine learning model if the correlation is harder to learn than the actual class itself. To this end, efforts such as that of [17] focused on learned spurious correlations, by investigating validation samples that are misclassified by a model of interest. However, this error-based approach assumes that the validation set is in some sense exhaustive and that the validation samples are already able to expose the SCs learned by the model, which may be true for subpopulation shift setups, but hardly guaranteed in a more general one.

Different from prior work, we aim to overcome the limitations of data and error analysis, revealing spurious correlations learned by machine models even in scenarios in which the training and validation sets do not present counterexamples able to expose the model's flaws. To this end we propose **WASP**, a **W**eight-space **A**pproach to detecting learned **Sp**urious correlations. Our overall framework is illustrated in Fig. 2 and further detailed in Sec. 3. We leverage the topology of foundation models such as CLIP and mGTE and observe that during the process of training, the weights of the final classification layer drift away from the textual representation of their associated class, towards identifying and prioritizing representations of spurious attributes. We propose a scoring system based on the model's embedding-space structure to extract concepts that factor in the activation of class neurons and delineate the highest-ranking concepts that lie outside the semantic scope of the classes, as spurious correlation.

Our contributions can be summarized as follows:

1. We introduce **WASP**, a weight-space approach to detecting *learned spuriousness*, offering a departure from the current error or data analysis methodology.

2. We show that our approach surpasses all state-of-the-art methods for *identifying and naming spurious correlations* in a given dataset. First, in its capacity to enhance the robustness of zero-shot models, and second, in its applicability to scenarios lacking spurious correlation counterexamples. Furthermore, in addition to evaluating our method on the established image datasets (Waterbirds, CelebA), we also validate its effectiveness on textual data (CivilComments).

3. We show that our method is able to *expose previously untapped ImageNet-1k spurious correlations*, and further proceed to show that multiple state-of-the-art models are affected by them, highlighting their inability to formulate robust definitions for the targeted classes.

## 2  Background

Machine learning methods naturally capture relevant factors needed to solve a task. However, models might also capture shortcuts [11], as correlations between non-essential features of the inputs and the label. These shortcuts represent spurious correlations, that don't hold in a more general setup (*e.g.* using water features to classify waterbirds instead of focusing on the birds' features), and should not be used for reliable generalization outside the training distribution, as they often lead to degraded performance [34, 3, 13].

**SCs from error analysis**  Approaches like **B2T** [17] rely exclusively on the validation samples, identifying which correlations between concepts are more prevalent in the misclassified examples. To catch SCs, B2T needs samples that oppose the strong correlations in the training set, thus leading to misclassification. To circumvent this limitation, **DrML** [56] manually builds a list of texts containing classes and concepts associations, that could potentially underline an SC. It forwards each such textual association through a classifier, learned on the image modality, and keeps as SCs the erroneously classified ones. In those approaches, the burden falls on the practitioners to come up with exhaustive samples or associations, failing to detect unexpected SCs. Differently, **WASP** focuses on analyzing the explicit learned weights of a model, covering all trainset samples. We thus extract the spuriously correlated concepts directly from the weights, bypassing the need for an exhaustive validation set or correlations candidates.

**SCs from train data analysis**  In **SpLiCE** [4] each image is decomposed into high-level textual concepts, searching next for concepts that are frequent for a certain class, but not for the others. **LG** [58] relies on LLMs to propose concepts potentially correlated with each class, using image captions. Next, it uses CLIP [35] to estimate a class-specificity score for each concept, and highly scored concepts for a class w.r.t. the others are considered SCs. These methods focus on the concepts' occurrences per class, making them prone to missing low-frequency concepts, as their presence can be drowned when averaging scores over a large dataset. Moreover, the SCs found through data analysis could be harder to learn than the class itself, so they are not necessarily imprinted in the model. In contrast, learned SCs (including error analysis revealed ones) must always be addressed, as they are, by definition, proven to impact the classifier. For this reason, **WASP** targets learned SCs by looking directly at the impact of the training set upon the model's weights.

**Manual interpretation of correlations**  The method introduced in [41] finds spurious features learned by a model, but it requires humans to manually annotate whether an image region is causal or not for a class. While this ensures a higher quality of the annotations, it also poses problems of scalability to large datasets. In contrast, **WASP** works fully automated, at scale, identifying SCs for each class in ImageNet-1k.

**SCs from subpopulation shift setup**  Other previous works [33, 23, 2, 54] have focused on SC identification strictly within the context of subpopulation shifts. The particularity of this setup is that the training and validation sets always contain subsets of samples that oppose the strong spurious correlations of the dataset. Most of these methods [33, 23, 54] focus on first learning a strongly biased classifier and then either separate the samples of each class into two groups [33, 54] (one containing correctly classified examples and the other containing misclassified ones), or place higher weights on hard samples [23], in order to balance the dataset. **CoBalT** [2] on the other hand uses an

unsupervised method for object recognition and then samples the dataset examples such that all object types are uniformly distributed in each class. Their result contains heatmaps overlays on images, which can offer insights to guide further manual SCs identification. Some of the most commonly used datasets in this setup are Waterbirds [37], CelebA [26] and CivilComments [5].

**Preventing the learning of SCs** As the statistical correlation of attributes and classes lies at the root of learning SCs, breaking this correlation is an accessible way of preventing their learning. Assuming that the training set features all combination of classes and attributes, this can be achieved by balancing all the existing groups of samples, as defined by the intersection of class and attribute labels. **GroupDRO** [37] uses group-specific weights that are dynamically updated during training to balance them, and it is the approach most commonly taken by works that simply find dataset partitions [33, 54], and also by works that name the SCs, and then obtain pseudo-labels for those attributes [17, 58]. To judge the robustness of a classifier, the **worst-group accuracy** metric is employed, which computes the accuracy on each individual group of samples and then reports their minimum. The worst-group accuracy of GroupDRO with ground truth attribute labels is usually viewed by the previously mentioned works as an upper bound on the performance that can be obtained.

## 3 Our Method

For a standard classification task, we aim to identify spurious correlations learned by a model through training on a new dataset. By **dataset's SCs** we refer to class-independent concepts, whose presence in the samples greatly affects the class label distribution. By **concepts** we refer to words or expressions with a well-defined semantic content. Through **SCs learned by a classifier** $f_\theta$ we refer to concepts causally unrelated with a class, whose presence in the input significantly changes the distribution of class probabilities predicted by $f_\theta$. We further classify them as positively correlated concepts w.r.t. to a class $k$, if their presence in the input increases the probability of $f_\theta$ predicting the class $k$, and negatively correlated concepts if they decrease it.

**Setup** We start with a dataset of samples $(x_j, y_j) \in (\mathcal{X}, \mathcal{Y})$ and construct the set of concepts $c_i \in C_{all}$ in textual form, that are present in the training data. We use a foundation model $M$, capable of embedding both the input samples $x_j$ and the concepts $c_i$ in aligned representations in $\mathbb{R}^D$. The main steps of our method (also revealed in Fig. 2, and in more detail in the Alg. 3.1), are the following:

**Initialization** We train a linear layer on top of the embedding space from $M$. We initialize $w_k$, the weights of class $k$ in this layer, with the embedding of its corresponding class name, extracted by the model $M$:

$$w_k^0 = M(\text{class\_name}_k), k \in \overline{1, |K|}, \tag{1}$$

where $K$ is the list of class names.

**Step 1: Model training** We train the weights of the linear layer using ERM [48]. Through learning, the weights for each class $k$ in the linear layer naturally shift from their original initialization, $M(\text{class\_name}_k)$, towards $w_k^*$, as visually presented in Fig. 2.

**Step 2: SCs identification** Since the weights $w_k$ and concepts $c_i$ share the same embedding space, we continue by identifying which concepts $c_i$ are correlated with each class $k$, as follows:

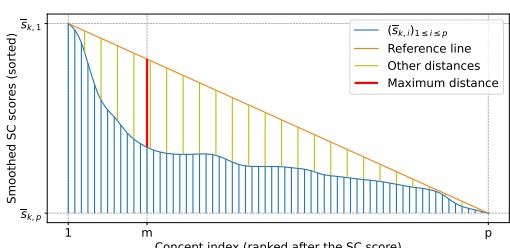

Figure 3: The maximum distance between the reference line and the smoothed scores gives the threshold for our cut-off heuristic.

**a. Filter out class-related concepts** After extracting $c_i$ concepts present in the dataset using existing tools, we filter out the concepts that are related to any actual class, leaving only concepts that are causally unrelated to all classes, which we call **class-neutral concepts**. We argue that only these concepts are proper candidates for SCs, as they are not required nor useful for the robust recognition of a class (*e.g.* a forest background is well correlated with species of landbirds, but we want to prevent the model from relying on this

correlation, which is causally unrelated w.r.t. the class definition). The exact pipeline and tools for processing the concepts (extraction and filtering) are detailed in Sec. 4.

**b. Rank class-neutral concepts** For each class-neural concept $c_i$ and class $k$, we rank the concepts based on their similarities with the learned class weights $w_k^*$, using the following positive-SC score:

$$s_{k,i}^+ = w_k^{*\top} M(c_i) - \min_{k' \in 1, |K|} w_{k'}^{*\top} M(c_i). \tag{2}$$

Intuitively, we want concepts similar to one class but not to all the others. Thus, for each class $k$, we select the concepts which starkly correlate to it, compared to all other classes. For the negatively correlated SCs, $s_{k,i}^-$ we use the dissimilarity score: $-w_k^{*\top} M(c_i)$.

**c. Dynamic thresholding** We next keep only the highest-ranked SCs of each class, using a *dynamic threshold* for the scores above, that allows us to automatically select the SCs for each class. To smooth the curve of scores, we apply a mean filter on top of the ranked concepts (window size $r$). We denote the scores obtained at this step with $(\overline{s}_{k,i})_{1 \leq i \leq p}$, with $p = q - r + 1$, where $q$ is the total number of filtered concepts. We then select the top $m_k$ ones, as positive SCs for class $k$, where the $m_k$ index is defined as:

$$m_k = \lfloor r/2 \rfloor + \arg\max_i \left( \overline{s}_{k,1} - i \frac{\overline{s}_{k,1} - \overline{s}_{k,p}}{p-1} - \overline{s}_{k,i} \right). \tag{3}$$

The intuition is that $m_k$ represents the index where the curve of smoothed scores, $\overline{s}_{k,i}$, deviates the most from the straight reversed diagonal line, connecting $\overline{s}_{k,1}$ and $\overline{s}_{k,p}$, like visually shown in Fig. 3.

### 3.1 Algorithm

We present the pseudocode of our proposed WASP approach in Alg. 1. We annotate the main steps presented in Sec. 3. At line 1, we initialize the class weights of our linear probing layer with the text embeddings of the class names, the zero-shot classification weights of the foundation model $M$. We then fine-tune the layer (line 2), on the given dataset. At lines 3-4, we filter the list of concepts and compute the embeddings for the remaining class-neutral concepts. The filtering can be performed by means of employing WordNet associations, Large Language Models, or both, as described in Sec. 4. With the embeddings of the class-neutral concepts at hand, we proceed to determine the SCs for each class. At line 5, we initialize our set of spuriously correlated concepts with an empty set, and we compute the similarities between each selected concept and each class at line 6. Next, for each class $k$, we compute the set of scores for all concepts w.r.t. this class and store it in $s_k$ (line 8). For each class-neutral concept, its ranking score for the current class is the difference between its similarity to the current class and the smallest similarity with a different class. Lines 9-11 formally implement the dynamic thresholding procedure, described in Sec. 3. Finally, we select the top concepts (above the computed threshold $m_k$) and store them as spuriously correlated concepts for class $k$ (lines 12-13).

## 4 Experimental Setup

**Foundation models (FM)** We used mGTE (gte-large-en-v1.5 [55]) for text embeddings in CivilComments [5], and OpenAI CLIP ViT-L/14 [35] for text and images otherwise.

**Concept extraction** For the image classification task, we first use the GIT-Large [51] captioning model (trained on MSCOCO [22]) to obtain descriptions of the dataset's images. Next, we extract concepts from the captions (or directly from the text samples for the text classification task), using YAKE [7] keyword extractor, taking the top 256 n-grams for $n = 3, 5$.

**Concepts filtering** We use Llama-3.1-8B-Instruct [10] to remove class instances from the concepts extracted at the previous step. We also apply a post-processing based on WordNet [30] to catch obvious class instances that the LLM might miss. For each class we specify a word used to search for synsets in the WordNet [30] hierarchy (*e.g. bird* for Waterbirds) and then remove individual words that match with any hyponym or hypernym of those synsets.

**Training** We train the linear layer on $L_2$ normalized embeddings extracted by the FM, using PyTorch's [32] AdamW [28] optimizer with a learning rate of $1e - 4$, weight decay of $1e - 5$ and batch size of $1024$. We use the cross entropy loss with balanced class weights as the objective. The weights of the layer are normalized after each update and we use CLIP's [35] temperature to scale the logits. We use the validation set's class-balanced accuracy for model selection and early stopping.

**Algorithm 1** WASP - Weight space Approach to detecting learned SPuriousness

**Input**: M - foundation model with associated text encoder; $(\mathcal{X}, \mathcal{Y})$ - Training set; $(\mathcal{X}_{val}, \mathcal{Y}_{val})$ - Validation set; K - list of class names; $C_{all}$ - list of all concepts; r - window for dynamic thresholding.
**Output**: Identified positive SCs: $\mathcal{B}$.

1: $\mathbf{W^0} \leftarrow M(K)$
2: $\mathbf{W} \leftarrow \text{ERM}\left(\mathbf{W^0}, M, (\mathcal{X}, \mathcal{Y}), (\mathcal{X}_{val}, \mathcal{Y}_{val})\right)$      ▷ 1. Model Training
3: $C \leftarrow \text{Filter}(C_{all})$      ▷ 2a. Filter concepts: LLM/WordNet
4: $\mathbf{C^*} \leftarrow M(C)$
5: $\mathcal{B} \leftarrow \emptyset$
6: $\mathbf{S} \leftarrow \mathbf{W}^\top \mathbf{C^*}$      ▷ 2b. Rank class-neutral concepts
7: **for** $k \in \overline{1, |K|}$ **do**
8:      $s_k = \left[\mathbf{S}_{k,j} - \min_{k' \in \overline{1,|K|}} \mathbf{S}_{k',j}, \text{ for all } j \in \overline{1, |C|}\right]$      ▷ 2c. Dynamic thresholding
9:      $\overline{s}_k = \text{mean\_pool}(\text{reversed}(\text{sorted}(s_k)), r)$
10:      $p = |C| - r + 1$
11:      $m_k = \lfloor r/2 \rfloor + \arg\max_i \left(\overline{s}_{k,1} - i\frac{\overline{s}_{k,1} - \overline{s}_{k,p}}{p-1} - \overline{s}_{k,i}\right)$
12:      $b_k = [s_{k,i} \mid i \leq m_k]$
13:      $\mathcal{B} = \mathcal{B} \cup (k, b_k)$      ▷ Positive SCs
14: **end for**

## 4.1 Datasets

**Waterbirds** [37] is a common dataset for generalization and mitigating spurious correlations. It is created from CUB [49], by grouping species of birds into two categories, *landbirds* and *waterbirds*, each one being spuriously correlated with the background, land, and water respectively.

**CelebA** [26] is a large-scale collection of celebrity images (over $200,000$), widely used in computer vision research. For generalization context, the setup [27] consists of using the *Blond_Hair* attribute as the class label and the *gender* as the spurious feature.

**CivilComments** [5] is a large collection of 1.8 million online user comments. This dataset is used employed in NLP bias and fairness research concerning different social and ethnical groups.

**ImageNet-1k** [8] is a larger-scale popular dataset for image classification (1000 classes, with approx. 1300 training samples and 50 validation samples per class).

Table 1: SC-enhanced zero-shot prompts. Following B2T [17], we explicitly introduce the SCs in the prompting scheme of the Foundational Models, leveraging that a more complete description of the image aids the zero-shot classification process. We note that using SCs identified by WASP significantly improves the worst group accuracy across all datasets (image and text modalities).

| Zero-shot | Waterbirds (Acc % ↑) | | CelebA (Acc % ↑) | | CivilComm (Acc % ↑) | |
|---|---|---|---|---|---|---|
| | Worst | Avg. | Worst | Avg. | Worst | Avg. |
| Basic | 35.2 | 84.2 | 72.8 | 87.7 | 33.1 | 80.2 |
| w B2T | 48.1 | 86.1 | 72.8 | 88.0 | - | - |
| w SpLiCE | 48.1 | 82.5 | 67.2 | 90.2 | - | - |
| w Lg | 46.1 | 85.9 | 50.6 | 87.2 | - | - |
| w **WASP** | **50.3** | 86.3 | **73.1** | 85.7 | **53.2** | 71.0 |

## 5 Evaluating spurious correlations

For a proper quantitative evaluation of our proposed SCs, beyond the subjectivity of the qualitative aspects, we combine the concepts identified by **WASP** with different components: in Sec. 5.1 we use SC-enhanced prompts for zero-shot classification with a FM; in Sec. 5.2 we evaluate using scenarios lacking spurious correlation counterexamples; and in Sec. 5.4 we generate samples exploiting the discovered concepts; An extended list of the extracted concepts can be found in Appx. F.

### 5.1 Spurious-aware zero-shot prompting

To further validate our identified spurious correlations, we follow [17] and evaluate them in the context of a zero-shot classification task. We augment the initial, class-oriented prompt with the identified concepts through a *minimal* intervention (*e.g.* 'a photo of a {cls} in the {concept}' (see Appx. G). For each class we create a prompt with each identified spurious correlation. When classifying an image, we take into account only the highest similarity among the prompts of a class

Table 2: Learning in the context of perfect spurious correlations. In the absence of samples that associate a class instance and concepts spuriously correlated with other classes, GroupDRO does not outperform the standard ERM. In contrast, our regularization based on the identified concepts consistently yields improvements (concerning worst group accuracy) over the considered baselines: ERM, GroupDRO, and the regularization with random causally unrelated concepts (obtained after the filtering in Step2a).

| Method | Waterbirds (Acc % ↑) | | CelebA (Acc % ↑) | | CivilComments (Acc % ↑) | |
|---|---|---|---|---|---|---|
| | Worst | Avg. | Worst | Avg. | Worst | Avg. |
| ERM [48] | 43.2±5.7 | 72.7±2.2 | 9.6±1.0 | 58.2±0.4 | 18.6±0.3 | 49.9±0.2 |
| GroupDRO [37] | 38.9±5.4 | 71.2±2.0 | 8.1±0.3 | 60.3±1.0 | 18.7±0.4 | 50.2±0.5 |
| Regularize w/ random SCs | 46.6±2.7 | 75.3±1.1 | 9.4±0.0 | 61.4±2.0 | 19.1±1.6 | 50.8±0.9 |
| Regularize w/ Lg's SCs [58] | 50.4±0.1 | 76.6±0.0 | 8.3±0.0 | 61.2±0.5 | - | - |
| Regularize w/ **WASP**'s SCs | **57.9**±0.3 | 79.8±0.1 | **10.4**±0.5 | 62.0±1.8 | **31.3**±0.7 | 57.5±0.4 |

(zero-shot with max-pooling over prompts). See in Tab. 1 how the SCs revealed by our method improve the worst group accuracy over the initial zero-shot baseline and other state-of-the-art solutions, in all the tested datasets. This highlights the relevance of the SCs automatically extracted by WASP. See more ablation experiments in Appx. H.

## 5.2 Training in a Fully Spurious Setup

We explore here an extreme setup, featuring no spurious correlation counterexample. In real world, this might be the case for most of the SCs, since usually this kind of correlations are generated by decisions in dataset acquisitions and filtering. Removing the minority groups from common robustness datasets, renders GroupDRO-like approaches completely ineffective, as their performance at best only matches the standard ERM (see Tab. 2). We use the employed SCs to impose a regularization upon the trained linear probes, learned on top of frozen embeddings from the FM. Intuitively, we constraint the weights to be equally distanced from the identified SCs. We formulate this as an MSE between the similarity of class weight $w_k$ with an SC $b$ and the average similarity of all class weights with $b$, these terms being scaled by $\tau$ (CLIP's temperature):

$$\mathcal{L}_{reg}(b) = \frac{\tau^2}{N} \sum_{k=1}^{N} \left[ w_k^\top M(b) - sg\left( \frac{1}{N} \sum_{j=1}^{N} w_j^\top M(b) \right) \right]^2, \quad (4)$$

with $sg$ being the stop gradient operator. The final loss is $\mathcal{L} = \mathcal{L}_{ERM} + \alpha \frac{1}{|\mathcal{B}|} \sum_{b \in \mathcal{B}} \mathcal{L}_{reg}(b)$, where $\mathcal{B}$ is the set of selected concepts and $\alpha = 0.1$. In Tab. 2, we present the results of linear probing with this loss, in the previously mentioned scenario, with no SC counterexamples, just 100% correlations between chosen tuples of concepts and classes. Through SC regularization, the learned classification weights are less reliant on the revealed SCs. The improvement in worst group accuracy shows the new representations are more robust and better aligned with the classification task, underling that WASP identifies concepts that are truly spuriously correlated with the classes. For a better comparison, we replicate the SC identification process of Lg [58] in this scenario.

## 5.3 Qualitative examples

We present in Tab. 3 the concepts identified as spuriously correlated with each class by WASP and competitors. Notice how our method discovers many new concepts (in blue) when compared with others. This is because our approach is fundamentally different, as it relies on the decision-making process of the model being investigated, diverging from current techniques oriented to validation set errors (B2T), or others that do data analysis over frozen concepts (SpLiCE, Lg). For CelebA-*blonde hair*, B2T and WASP do not find any SCs. This turn out to be an appropriate decision, since the presence of the feminine features do not incline the model towards one class or the other. See an exhaustive list of SCs revealed by WASP (ImageNet-1k included) in Appx. F.

Table 3: Qualitative SCs examples, extracted on Waterbirds, CelebA and CivilComments datasets. See in red concepts that are off-topic, person names, or too related to the semantic content of the class, and in blue new concepts, that were not identified before. WASP, w.r.t. others, focus on learned SCs, discovering many new spuriously correlated concepts (and expressions, marked with ...).

| | Waterbirds | | CelebA | | CivilComments | |
| | *landbird* | *waterbird* | *blonde hair* | *non-blonde hair* | *offensive* | *non-offensive* |
|---|---|---|---|---|---|---|
| B2T [17] | forest, woods, tree, branch | ocean, beach, surfer, boat, dock, water, lake | - | man, male | - | - |
| SpLiCE [4] | bamboo, perched, rainforest | flying | hairstyles, dolly, turban, actress, tennis, beard | hairstyles, visor, amy, kate, fielder, cuff, rapper, cyclist | - | - |
| Lg [58] | forest, woods, rainforest, tree branch, tree | beach, lake, water, seagull, pond | ...blonde hair, actress, model, woman long hair | man..., sunglasses, young man, black hair, actor | - | - |
| **WASP** (ours) | forest..., bamboo..., ground, field, log, grass..., tree | swimming..., water, lake, flying..., boat, lifeguard, pond | - | hat..., man..., actor, person, dark, large, shirt | hypocrisy, troll, solly, hate | allowing, work, made, talk |

Table 4: Results for three positively correlated SCs found using WASP for CLIP ViT-L/14 fine-tuned on ImageNet. We evaluate the model's capability to recognize a depicted (correct) class before and after the introduction of an identified concept in the image. For each prompt, 1000 images are generated using FLUX.1-dev. We observe throughout all considered scenarios, a significant drop in the model's capacity to identify the correct class when the selected concept is involved and a large increase in the likelihood of having the induced class predicted even though it is not illustrated.

| Correct Class | Exploited SC (Induced Class) | Prompt | Samples Predicted As (%) | |
| | | | Correct Class | Induced Class |
|---|---|---|---|---|
| **peafowl** | **firemen** (**fire truck**) | • a photo of a **peafowl** 
 • **firemen** and a **peafowl** | 100.0 
 5.3 (**-94.7**) | 0.0 
 93.4 (**+93.4**) |
| **Mexican hairless dog** | **reading a newspaper** (**crossword**) | • a photo of a **Mexican hairless dog** 
 • a man **reading a newspaper** in a chair with a **Mexican hairless dog** in his lap | 47.5 
 0.9 (**-46.6**) | 0.0 
 36.6 (**+36.6**) |
| **Bernese Mountain Dog** | **shrimp** (**American lobster**) | • a photo of a **Bernese Mountain Dog** 
 • **shrimp** and pasta near a **Bernese Mountain Dog** | 99.8 
 10.6 (**-89.2**) | 0.0 
 37.2 (**+37.2**) |

## 5.4 ImageNet Spurious Correlations

In the previous subsections we have shown that our proposed method exhibits the desired behavior in the controlled setups popular within the subpopulation shift literature aimed at identifying and preventing spurious correlations. We have also shown that the SCs found by our method aid in improving the zero-shot performance of CLIP, outperforming existing approaches, and that WASP is applicable to situations which lie outside the scope of existing approaches, namely: (i) it is applicable in scenarios in which the training data features 100% spuriously correlated samples, with no counterexamples to point out the SCs, and (ii) it is applicable to both image and text datasets.

Within this subsection, we venture even further and apply our method in an uncontrolled, general setup. Specifically, we employ WASP to point out spurious correlations plaguing the decision-making process of OpenAI's CLIP ViT-L/14 fine-tuned on ImageNet. Within the ImageNet setup, the current state-of-the-art approach, B2T [17], points out the SCs learned by the model by analyzing the mistakes the model makes when evaluated on the validation set. Different from B2T [17], our approach does not rely on the validation data to provide counterexamples able to expose the SCs, and it is able to provide a list of SCs which exceeds the scope of the validation dataset. We provide extensive lists of SCs pointed out by our method in Appx. F. We note that most of the SCs pointed out by our method are previously untapped, opening up a new avenue for investigating ImageNet SCs.

We further invest the effort to generate and manually verify images in order to open up this avenue and showcase previously undiscovered flaws in state-of-the-art models. To this end, we employ a quantized version of FLUX.1-dev [20], and in order to validate the impact of the SCs, we prompt

Table 5: Accuracy of various convolutional and transformer-based models trained on ImageNet-1k, on the data generated for Tab. 4. As with Fig. 1, we note that the performance of these models is significantly affected, even though the correct class is illustrated right in front and center while the predicted class is absent from the generated images. An exhaustive list is presented in Appx. I.

| | Prompt employed (correct class highlighted in **bold and blue**, SC in yellow) | | | |
|---|---|---|---|---|
| Model | a photo of a **peafowl** | firemen and a **peafowl** | a photo of a **Bernese Mountain Dog** | shrimp and pasta near a **Bernese Mountain Dog** |
| alexnet [19] | 100.0 | 4.6 (**-95.4**) | 96.2 | 23.3 (**-72.9**) |
| efficientnet_b1 [45] | 100.0 | 42.6 (**-57.4**) | 88.1 | 67.1 (**-21.0**) |
| regnet_x_32gf [36] | 100.0 | 66.1 (**-33.9**) | 85.9 | 46.0 (**-39.9**) |
| resnet50 [12] | 100.0 | 30.1 (**-69.9**) | 73.9 | 54.5 (**-19.4**) |
| resnext101_32x8d [52] | 100.0 | 66.6 (**-33.4**) | 84.7 | 61.2 (**-23.5**) |
| squeezenet1_1 [16] | 100.0 | 13.8 (**-86.2**) | 91.2 | 46.1 (**-45.1**) |
| swin_b [24] | 100.0 | 81.5 (**-18.5**) | 95.2 | 72.6 (**-22.6**) |
| vgg19_bn [40] | 100.0 | 35.9 (**-64.1**) | 83.1 | 46.2 (**-36.9**) |
| vit_l_16 [9] | 100.0 | 55.9 (**-44.1**) | 95.3 | 76.0 (**-19.3**) |
| wide_resnet50_2 [53] | 100.0 | 60.6 (**-39.4**) | 95.7 | 63.9 (**-31.8**) |

the generative model to depict a chosen (correct) class alongside a non-ImageNet object that we identified as a positive SC for a different class.

The validation process is presented for three distinct scenarios in Tab. 4. Each scenario is defined by a correct class that is illustrated in the image, a concept (object, property, or activity) that is not causally tied to any class, and an absent class *induced* through the presence of the concept. We expect the classifier to predict this absent class, based on our scoring. We measure the impact of the concept by comparing the model's ability to predict the correct class before and after its introduction. We generate and manually ensure the compliance of 1000 images for each scenario and we evaluate the model both in terms of accuracy and in terms of the frequency with which it predicts the induced class. Throughout all considered scenarios, we observe a significant drop in the model's capacity to identify the correct class when the SC factor is involved, with an increased likelihood of having the induced class predicted, even though it is not illustrated in any way, shape or form in the image.

Within the same context, we present a series of qualitative examples in Fig. 1. We emphasize that, even though throughout most of these samples, a single ImageNet-1k class is clearly depicted, the model chooses to ignore it and label the image as a completely different class, not illustrated at all in the image, solely based on the presence of a non-ImageNet object. We underline, by means of the results presented in Tab. 4, that the model is not fooled by artifacts in the generated images to predict randomly. We test the performance of the models on images featuring the correct class, without added objects. We observe this way that the model's performance on the generated data is on par with the original performance of the model on these classes, validating that the generated images are not out of distribution. Furthermore, we show that the rate at which the induced class is predicted increases significantly.

The model at hand is generally considered to be a robust state-of-the-art model, benefiting from ample pre-training. We emphasize through our experiment that even under these reassuring circumstances, critical reasoning flaws can make their way through, in a production-ready model, undetected by validating and examining the model's performance on held-out data. We further proceeded to examine the impact of the SCs for which we have generated data on an exhaustive set of ImageNet-1k state-of-the-art models. We present the entire set of results in Appx. I, and the results for a selection of these models in Tab. 5. We emphasize that, even large transformer models, such as ViT-L/16 are heavily influenced by learned SCs. These results thus showcase the generality of our findings and the major impact that SCs silently had on state-of-the-art models.

# 6 Conclusions

We introduce **WASP**, a method that automatically identifies SCs through a weight-space analysis that can be scaled to large datasets. We have evaluated our approach on existing benchmarks, showcasing the relevancy of the proposed SCs, while proving that, different from existing work: (i) our method is applicable to both text and image datasets and that (ii) it is applicable in scenarios featuring fully spuriously correlated samples. Furthermore, using our method we have discovered previously untapped ImageNet SCs and showed that they affect multiple state-of-the-art models.

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

# Appendix

## A    Broader impact statement

By systematically detecting a wide spectrum of spuriously correlated concepts, our work stands to enhance the reliability and trustworthiness of AI-driven decisions across various real-world contexts. WASP could help researchers and developers to address unintended consequences that arise when models latch onto misleading data associations, drawing attention to critical responsibilities tied to deploying AI at scale.

## B    Software and data

We attach the PyTorch [32] implementation of WASP as supplementary material, including a *README.md* file to explain the code, which we will make publicly available. The datasets and pre-trained models used for WASP are already public.

## C    Limitations

Some limitations of WASP:

- **Concepts vs Input features as SCs**. The learned SCs can be described by our method in relation with the predefined (large set of) concepts, but not directly w.r.t. the input features (*e.g.* GradCAM [39] like methods).

- **Captioning model used for extracting concepts**. These models usually do not extract all the details in the images, so relying on them limits the concept space, that limits further discovering all SCs from the original images.

- **SCs from a dataset (only) through the lens of a Foundation Model**. While the Foundation Models are usually very robust ones, some SCs (specially those related to low-level - pixel-level - information) can disappear in the high-semantic embedding space of the foundation model, making it impossible for WASP to detect such SCs.

- **Relying on known hierarchies of concepts** The method also relies on known hierarchies of concepts (like WordNet) to filter out concepts related to the desired class. These hierarchies and the relations they provide thus limit the type of filtering that we can ensure.

## D    Additional related work

**Fairness** It is important to note that the proposed method can be utilized to evaluate the fairness of a given dataset and that we do conduct benchmarking on the CivilComments dataset, which encompasses racial and religious concerns. However, it is critical to emphasize that our approach is neither designed to measure nor address issues of fairness. Instead, our method is specifically developed to examine whether a given dataset imparts a clear definition of the featured classes to a model — namely, whether classifiers learn spurious correlations and confound class features with environmental features. Accordingly, our work is situated within the literature on subpopulation shift setups and we assess the quality of our proposed approach within this framework. Evaluating our approach on fairness benchmarks lies outside the scope of the current study, but may constitute a subject for subsequent research.

**Concept Bottleneck Models** Another approach to detecting spurious correlations would be to use models that are interpretable by design, such as Concept Bottleneck Models (CBMs) [18]. CBMs feature a special layer where each neuron's activations signals the presence or absence of a specific concept within the input sample. This makes it easier to see which concepts are used by the model down the line and also allows a user to filter out the concepts that he may consider as irrelevant for the task at hand. On the downside, CBMs, as proposed by Koh et al. [18], require a human expert to define the set of relevant concepts for each task and also concept-level annotations in a dataset in order to train the concept extraction layer. To circumvent these limitation, Oikarinen et al. [31] use concepts proposed by GPT-3 and then obtain pseudo-labels for those concepts using a CLIP

Table 6: Class names and prompts used in the zero-shot classification task.

|  | Waterbirds | CelebA | CivilComments |
|---|---|---|---|
| class names | waterbird
landbird | non-blonde hair
blonde hair | non-offensive
offensive |
| zero-shot prompt | a photo of a {cls} | a photo of a person with {cls} | {cls} |
| SC-enhanced prompt | a photo of a {cls}
in the {SC} | a photo of a {SC}
with {cls} | a/an {cls} comment
about {SC} |

model. This intervention of CBMs on a models's architecture constrains its reasoning space down to the set of predetermined concepts, yielding, compared to unaltered models, drops in accuracy of up to 4.97%, as reported by Oikarinen et al. [31] in Table 2. Different from this line of works, we never constrain the model in any way, shape or form. What we aim to uncover are SCs learned by general state-of-the-art models used in the industry, which are not explainable by design. Overall both approaches offer a different tradeoff between explainability and expressivity.

# E   Loss correlation with presence of spuriously correlated concepts

In this experiment, we look at the correlations between the loss values and the concept-to-sample similarities. We compare basic ERM with GroupDRO, applied on groups, that are obtained based on our revealed SCs (and further grouped using the B2T [17] partitioning strategy).

See in Fig. 4 how for GroupDRO, the loss-to-similarity correlation significantly decrease, revealing that the model is less prone to make mistakes on the samples containing SCs. The results show a reduction in correlation scores across all SCs, demonstrating that the revealed groups are relevant to the dataset's underlying distribution, and can be effectively utilized with specific algorithms to mitigate the model's dependence on spurious correlations. Fig. 4 shows the Pearson correlation scores after an epoch of training on Waterbirds, on a subset of all SCs.

# F   SCs and top class-neutral concepts

We present the exhaustive list of SCs found for the Waterbirds (Tab. 11 & 12), CelebA (Tab. 13) and CivilComments (Tab. 14 & 15) datasets. We also present top class-neutral concepts for ImageNet classes. Concept filtering on ImageNet was performed using WordNet relationships alone, without the intervention of a Large Language Model. Accounting for the size of the dataset, we will publish the available data on our repository upon acceptance, and we will restrict the presentation within the context of the current format to a few classes for illustrative purposes in Tab. 24 -17.

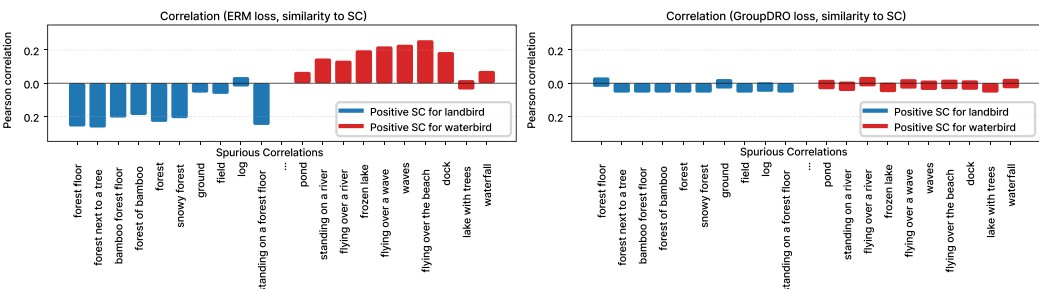

Figure 4: Correlation(sample_loss, sample_to_bias similarity) under ERM/GDRO after one epoch of training on Waterbirds. Loss correlation w/ biases, ERM vs GroupDRO using groups created with the B2T partitioning method. It can be seen that, when training with ERM, loss value is highly correlated with the biases. In contrast, GroupDRO reduces the correlations, intuitively showing that biases discovered with our method are closely related to the ground truth groups of the dataset, being used as shortcuts by the model unless mitigated.

## G  Zero-Shot Prompts

In Tab. 6, we structured the class names used for initializing the initial weights of the linear layer, along with the prompt templates employed in the zero-shot classification experiments discussed in Sec. 5.1.

## H  Ablations

We validate several WASP decisions in Tab. 7, for zero-shot classification task, using prompts enhanced with SCs. The number of SCs per class turns out to be very important, taking too many adds noise to the prompts and lowers the performance. Nevertheless, dynamically choosing the threshold, as described in Sec. 3-Step2c., proves to be a good strategy for adapting the cut-off across classes. Following prior observations regarding the modalities gap between text and image embedding space [21, 57], we subtract half of the gap from the embeddings and re-normalize them, ending in marginally lower performance w.r.t. not addressing the gap.

Table 7: Ablation. Following the zero-shot SC-augmented prompting setup, we variate the cut-off threshold for considering causally-unrelated concepts as spuriously correlated and also try to address the text-image modality gap.

| | Waterbirds (Acc % ↑) | | CelebA (Acc % ↑) | | CivilComm (Acc % ↑) | |
|---|---|---|---|---|---|---|
| Variations | Worst | Avg. | Worst | Avg. | Worst | Avg. |
| *top-30 candidates* | 46.3 | 86.1 | 66.2 | 86.4 | 46.9 | 71.1 |
| *top-20% candidates* | 46.1 | 86.0 | 64.8 | 86.7 | 48.8 | 66.2 |
| *modality gap: closed* | 48.7 | 85.9 | 72.7 | 85.3 | - | - |
| **WASP** *dynamic threshold* * modality gap: open* | **50.3** | 86.3 | **73.1** | 85.7 | **53.2** | 71.0 |

## I  Results of state-of-the-art models

We provide results using the same data and experimental setup used for Tables 4 and 5, for an exhaustive list of ImageNet classifiers, in Tables 8, 9 and 10. Pre-trained models together with their respective weight sets are employed from the `torchvision` package.

Table 8: Accuracy of various convolutional and transformer-based models trained on ImageNet-1k, on the data generated for Tab. 4. As with Fig. 2 and Fig. 1, we note that the performance of these models is significantly affected, even though the correct class is clearly illustrated right in front and center while, and the predicted class is absent from the generated images.

| | Prompt employed (correct class highlighted in bold, SC in red) | | | |
| Model - Weights | a photo of a **peafowl** | firemen and a **peafowl** | a photo of a **Bernese Mountain Dog** | shrimp and pasta near a **Bernese Mountain Dog** |
| --- | --- | --- | --- | --- |
| alexnet - V1 [19] | 100.0 | 4.6 | 96.2 | 23.3 |
| convnext_tiny - V1 [25] | 100.0 | 77.2 | 94.2 | 69.8 |
| convnext_small - V1 [25] | 100.0 | 92.9 | 96.3 | 78.3 |
| convnext_base - V1 [25] | 100.0 | 83.5 | 99.3 | 78.0 |
| convnext_large - V1 [25] | 100.0 | 88.2 | 99.6 | 81.9 |
| densenet121 - V1 [15] | 100.0 | 52.3 | 93.1 | 74.8 |
| densenet161 - V1 [15] | 100.0 | 52.8 | 84.0 | 51.6 |
| densenet201 - V1 [15] | 100.0 | 49.7 | 86.1 | 80.8 |
| efficientnet_b0 - V1 [45] | 100.0 | 64.5 | 99.2 | 92.9 |
| efficientnet_b1 - V1 [45] | 100.0 | 42.6 | 88.1 | 67.1 |
| efficientnet_b1 - V2 [45] | 100.0 | 84.2 | 99.9 | 69.5 |
| efficientnet_b2 - V1 [45] | 100.0 | 61.7 | 99.6 | 82.4 |
| efficientnet_b3 - V1 [45] | 100.0 | 89.1 | 99.1 | 92.8 |
| efficientnet_b4 - V1 [45] | 100.0 | 94.4 | 99.9 | 72.7 |
| efficientnet_b5 - V1 [45] | 100.0 | 82.9 | 99.4 | 86.3 |
| efficientnet_b6 - V1 [45] | 100.0 | 91.2 | 100.0 | 95.6 |
| efficientnet_b7 - V1 [45] | 100.0 | 88.3 | 99.9 | 90.3 |
| efficientnet_v2_s - V1 [46] | 100.0 | 98.3 | 99.2 | 88.1 |
| efficientnet_v2_m - V1 [46] | 100.0 | 95.1 | 99.8 | 94.3 |
| efficientnet_v2_l - V1 [46] | 100.0 | 96.0 | 99.1 | 83.6 |
| googlenet - V1 [42] | 100.0 | 45.4 | 95.2 | 57.0 |
| inception_v3 - V1 [43] | 100.0 | 82.2 | 98.9 | 80.8 |
| maxvit_t - V1 [47] | 100.0 | 91.7 | 99.7 | 85.3 |
| mnasnet0_5 - V1 [44] | 100.0 | 23.3 | 96.7 | 60.8 |
| mnasnet0_75 - V1 [44] | 100.0 | 36.9 | 98.4 | 71.1 |
| mnasnet1_0 - V1 [44] | 100.0 | 32.0 | 89.4 | 74.9 |
| mnasnet1_3 - V1 [44] | 100.0 | 63.9 | 91.7 | 75.6 |
| mobilenet_v2 - V1 [38] | 100.0 | 26.2 | 84.0 | 45.0 |
| mobilenet_v2 - V2 [38] | 100.0 | 54.1 | 98.6 | 62.8 |
| mobilenet_v3_small - V1 [14] | 100.0 | 16.6 | 91.0 | 30.3 |
| mobilenet_v3_large - V1 [14] | 100.0 | 25.4 | 94.2 | 23.1 |
| mobilenet_v3_large - V2 [14] | 100.0 | 56.0 | 96.6 | 73.7 |

Table 9: Accuracy of various convolutional and transformer-based models trained on ImageNet-1k, on the data generated for Tab. 4. As with Fig. 1, we note that the performance of these models is significantly affected, even though the correct class is clearly illustrated right in front and center while, and the predicted class is absent from the generated images.

| Model - Weights | Prompt employed (correct class highlighted in bold, SC in red) | | | |
|---|---|---|---|---|
| | a photo of a **peafowl** | firemen and a **peafowl** | a photo of a **Bernese Mountain Dog** | shrimp and pasta near a **Bernese Mountain Dog** |
| regnet_y_400mf - V1 [36] | 100.0 | 35.6 | 74.7 | 34.7 |
| regnet_y_400mf - V2 [36] | 100.0 | 72.1 | 97.7 | 87.9 |
| regnet_y_800mf - V1 [36] | 100.0 | 22.7 | 94.8 | 54.4 |
| regnet_y_800mf - V2 [36] | 100.0 | 81.7 | 98.9 | 88.0 |
| regnet_y_1_6gf - V1 [36] | 100.0 | 47.0 | 96.4 | 71.7 |
| regnet_y_1_6gf - V2 [36] | 100.0 | 88.2 | 96.4 | 74.1 |
| regnet_y_3_2gf - V1 [36] | 100.0 | 33.5 | 99.4 | 90.9 |
| regnet_y_3_2gf - V2 [36] | 100.0 | 94.6 | 98.4 | 75.4 |
| regnet_y_8gf - V1 [36] | 100.0 | 58.3 | 71.3 | 54.8 |
| regnet_y_8gf - V2 [36] | 100.0 | 98.2 | 96.8 | 67.5 |
| regnet_y_16gf - V1 [36] | 100.0 | 86.7 | 97.8 | 49.8 |
| regnet_y_16gf - V2 [36] | 100.0 | 98.7 | 91.8 | 83.5 |
| regnet_y_16gf - SWAG_E2E_V1 [36] | 100.0 | 99.6 | 97.7 | 78.2 |
| regnet_y_16gf - SWAG_LINEAR_V1 [36] | 100.0 | 78.3 | 100.0 | 90.7 |
| regnet_y_32gf - V1 [36] | 100.0 | 84.5 | 99.1 | 82.3 |
| regnet_y_32gf - V2 [36] | 100.0 | 98.0 | 99.5 | 79.6 |
| regnet_y_32gf - SWAG_E2E_V1 [36] | 100.0 | 99.6 | 93.5 | 71.5 |
| regnet_y_32gf - SWAG_LINEAR_V1 [36] | 100.0 | 97.2 | 100.0 | 85.6 |
| regnet_y_128gf - SWAG_E2E_V1 [36] | 100.0 | 99.6 | 54.4 | 67.8 |
| regnet_y_128gf - SWAG_LINEAR_V1 [36] | 100.0 | 90.8 | 99.9 | 96.9 |
| regnet_x_400mf - V1 [36] | 100.0 | 37.4 | 82.3 | 29.0 |
| regnet_x_400mf - V2 [36] | 100.0 | 50.7 | 99.5 | 81.7 |
| regnet_x_800mf - V1 [36] | 100.0 | 25.4 | 77.0 | 52.6 |
| regnet_x_800mf - V2 [36] | 100.0 | 75.5 | 97.1 | 71.4 |
| regnet_x_1_6gf - V1 [36] | 100.0 | 38.2 | 76.9 | 66.7 |
| regnet_x_1_6gf - V2 [36] | 100.0 | 82.2 | 99.2 | 88.1 |
| regnet_x_3_2gf - V1 [36] | 100.0 | 45.1 | 62.1 | 69.4 |
| regnet_x_3_2gf - V2 [36] | 100.0 | 83.4 | 99.6 | 88.3 |
| regnet_x_8gf - V1 [36] | 100.0 | 41.8 | 98.8 | 89.0 |
| regnet_x_8gf - V2 [36] | 100.0 | 93.5 | 99.2 | 81.6 |
| regnet_x_16gf - V1 [36] | 100.0 | 48.7 | 86.6 | 54.5 |
| regnet_x_16gf - V2 [36] | 100.0 | 93.4 | 97.9 | 88.1 |
| regnet_x_32gf - V1 [36] | 100.0 | 66.1 | 85.9 | 46.0 |
| regnet_x_32gf - V2 [36] | 100.0 | 97.4 | 99.6 | 83.7 |
| resnet18 - V1 [12] | 100.0 | 36.8 | 84.3 | 41.9 |
| resnet34 - V1 [12] | 100.0 | 32.9 | 54.1 | 26.1 |
| resnet50 - V1 [12] | 100.0 | 30.1 | 73.9 | 54.5 |
| resnet50 - V2 [12] | 100.0 | 80.1 | 99.7 | 88.4 |
| resnet101 - V1 [12] | 100.0 | 60.4 | 92.3 | 84.4 |
| resnet101 - V2 [12] | 100.0 | 93.4 | 98.8 | 87.4 |
| resnet152 - V1 [12] | 100.0 | 66.1 | 98.4 | 78.2 |
| resnet152 - V2 [12] | 100.0 | 93.1 | 98.5 | 92.5 |
| resnext50_32x4d - V1 [52] | 100.0 | 45.5 | 92.6 | 74.8 |
| resnext50_32x4d - V2 [52] | 100.0 | 80.3 | 98.5 | 88.0 |
| resnext101_32x8d - V1 [52] | 100.0 | 66.6 | 84.7 | 61.2 |
| resnext101_32x8d - V2 [52] | 100.0 | 90.3 | 99.4 | 85.2 |
| resnext101_64x4d - V1 [52] | 100.0 | 77.9 | 97.7 | 74.2 |

Table 10: Accuracy of various convolutional and transformer-based models trained on ImageNet-1k, on the data generated for Tab. 4. As with Fig. 2 and Fig. 1, we note that the performance of these models is significantly affected, even though the correct class is clearly illustrated right in front and center while, and the predicted class is absent from the generated images.

| Model - Weights | Prompt employed (correct class highlighted in bold, SC in red) | | | |
| --- | --- | --- | --- | --- |
| | a photo of a **peafowl** | firemen and a **peafowl** | a photo of a **Bernese Mountain Dog** | shrimp and pasta near a **Bernese Mountain Dog** |
| shufflenet_v2_x0_5 - V1 [29] | 100.0 | 23.8 | 36.9 | 21.0 |
| shufflenet_v2_x1_0 - V1 [29] | 99.8 | 30.4 | 72.1 | 55.9 |
| shufflenet_v2_x1_5 - V1 [29] | 100.0 | 41.2 | 97.9 | 52.5 |
| shufflenet_v2_x2_0 - V1 [29] | 100.0 | 61.4 | 99.3 | 64.5 |
| squeezenet1_0 - V1 [16] | 100.0 | 11.4 | 95.9 | 28.7 |
| squeezenet1_1 - V1 [16] | 100.0 | 13.8 | 91.2 | 46.1 |
| swin_t - V1 [24] | 100.0 | 72.7 | 96.9 | 81.6 |
| swin_s - V1 [24] | 100.0 | 74.3 | 99.3 | 81.4 |
| swin_b - V1 [24] | 100.0 | 81.5 | 95.2 | 72.6 |
| swin_v2_t - V1 [24] | 100.0 | 76.2 | 88.7 | 73.6 |
| swin_v2_s - V1 [24] | 100.0 | 85.7 | 90.7 | 74.4 |
| swin_v2_b - V1 [24] | 100.0 | 73.0 | 96.0 | 85.2 |
| vgg11 - V1 [40] | 100.0 | 9.9 | 96.6 | 60.7 |
| vgg11_bn - V1 [40] | 100.0 | 15.9 | 86.7 | 54.8 |
| vgg13 - V1 [40] | 100.0 | 5.9 | 97.7 | 68.1 |
| vgg13_bn - V1 [40] | 100.0 | 15.5 | 87.0 | 13.0 |
| vgg16 - V1 [40] | 100.0 | 12.2 | 93.7 | 66.0 |
| vgg16_bn - V1 [40] | 100.0 | 22.6 | 96.1 | 70.7 |
| vgg19 - V1 [40] | 100.0 | 26.6 | 98.5 | 40.8 |
| vgg19_bn - V1 [40] | 100.0 | 35.9 | 83.1 | 46.2 |
| vit_b_16 - V1 [9] | 100.0 | 85.4 | 97.1 | 75.2 |
| vit_b_16 - SWAG_E2E_V1 [9] | 100.0 | 88.9 | 95.8 | 59.1 |
| vit_b_16 - SWAG_LINEAR_V1 [9] | 100.0 | 79.2 | 99.9 | 84.6 |
| vit_b_32 - V1 [9] | 100.0 | 56.1 | 96.0 | 86.0 |
| vit_l_16 - V1 [9] | 100.0 | 55.9 | 95.3 | 76.0 |
| vit_l_16 - SWAG_E2E_V1 [9] | 100.0 | 92.1 | 100.0 | 93.3 |
| vit_l_16 - SWAG_LINEAR_V1 [9] | 100.0 | 97.4 | 100.0 | 72.5 |
| vit_l_32 - V1 [9] | 100.0 | 54.0 | 96.9 | 82.9 |
| vit_h_14 - SWAG_E2E_V1 [9] | 100.0 | 99.4 | 98.5 | 86.4 |
| vit_h_14 - SWAG_LINEAR_V1 [9] | 100.0 | 99.7 | 100.0 | 96.1 |
| wide_resnet50_2 - V1 [53] | 100.0 | 60.6 | 95.7 | 63.9 |
| wide_resnet50_2 - V2 [53] | 100.0 | 83.9 | 99.5 | 85.4 |
| wide_resnet101_2 - V1 [53] | 100.0 | 69.3 | 84.1 | 72.8 |
| wide_resnet101_2 - V2 [53] | 100.0 | 91.0 | 98.8 | 84.9 |

Table 11: Top Waterbirds class-neutral concepts for "landbird".

| Landbird | Score |
|---|---|
| forest floor | 0.055562317 |
| forest next to a tree | 0.053587496 |
| bamboo forest floor | 0.05134508 |
| forest of bamboo | 0.04781133 |
| forest | 0.047080815 |
| snowy forest | 0.044688106 |
| ground | 0.043406844 |
| field | 0.043168187 |
| log | 0.043052554 |
| standing on a forest floor | 0.041143 |
| grass covered | 0.040526748 |
| tree branch in a forest | 0.039670765 |
| forest with trees | 0.03949821 |
| tree in a forest | 0.039123535 |
| bamboo forest | 0.03876221 |
| front of bamboo | 0.037381053 |
| mountain | 0.036155403 |
| forest of trees | 0.03600967 |
| flying through a forest | 0.035692394 |
| platform | 0.03565806 |
| standing in a forest | 0.034528017 |
| hill | 0.03341371 |

Table 12: Top Waterbirds class-neutral concepts for "waterbird".

| Waterbird | Score |
|---|---|
| swimming in the water | 0.11482495 |
| water lily | 0.10905403 |
| boat in the water | 0.1066975 |
| floating in the water | 0.106155455 |
| water | 0.106134474 |
| flying over the water | 0.10561061 |
| standing in the water | 0.10444009 |
| sitting in the water | 0.103776515 |
| body of water | 0.09977633 |
| water in front | 0.0902465 |
| standing in water | 0.086544394 |
| water and one | 0.08424729 |
| swimming | 0.07818574 |
| standing on a lake | 0.06565446 |
| flying over the ocean | 0.06509364 |
| flying over a pond | 0.06463468 |
| boats | 0.061231434 |
| lifeguard | 0.06122452 |
| flying over a lake | 0.060126305 |
| boat | 0.0571931 |
| pond | 0.053261578 |

Table 13: Top CelebA class-neutral concepts for "non-blonde".

| Non-Blonde | Score |
|---|---|
| hat on and a blue | 0.13952243 |
| hat on and a man | 0.13853341 |
| man in the hat | 0.13803285 |
| man who made | 0.13307464 |
| man behind | 0.13247031 |
| man with the hat | 0.13186401 |
| man is getting | 0.12861347 |
| actor | 0.12726557 |
| dark | 0.12713176 |
| man in a blue | 0.1269682 |
| person | 0.12541258 |
| man in the blue | 0.124844134 |
| man is not a man | 0.12308431 |
| man | 0.12290484 |
| large | 0.1228559 |
| shirt on in a dark | 0.12170941 |
| hat | 0.121646166 |
| close | 0.12146461 |
| man with the blue | 0.12136656 |
| man face | 0.12130207 |

Table 14: Top CivilComments class-neutral concepts for "non-offensive".

| Non-offensive | Score |
| --- | --- |
| allowing | 0.07341421 |
| work | 0.06982881 |
| made | 0.069063246 |
| talk | 0.06858361 |
| none are needed | 0.067236125 |
| check | 0.06664443 |
| helping keep the present | 0.06531584 |
| policy | 0.06339955 |
| campaign | 0.06333798 |
| involved in the first place | 0.063222766 |
| Cottage | 0.063149124 |
| IDEA | 0.06310266 |
| stories | 0.0625782 |
| job | 0.06236595 |
| allowed | 0.062137783 |
| latest news about the origin | 0.062061936 |
| giving others who have experienced | 0.061925888 |
| proposed | 0.061897278 |
| one purpose | 0.06122935 |
| starting | 0.061154723 |
| small | 0.061071455 |
| question | 0.060854554 |
| practice | 0.060740173 |
| raised | 0.060681045 |
| entering | 0.060585797 |
| registered | 0.060475767 |
| beliefs | 0.060165346 |
| accept that they are promoting | 0.060070753 |
| Security | 0.059328556 |
| new | 0.059324086 |
| subject | 0.058983028 |
| close | 0.058632135 |
| views | 0.058573127 |
| Hold | 0.058341324 |
| reality for a change | 0.058261245 |
| built at that parish | 0.057885766 |
| rest | 0.057804525 |
| historic | 0.057656527 |
| concept | 0.057422698 |
| people | 0.057151675 |
| passage seems to in reflection | 0.05699992 |
| attempt | 0.056797385 |

Table 15: Top CivilComments class-neutral concepts for "offensive".

| "Offensive" – Top Concepts | Score |
| --- | --- |
| hypocrisy | 0.046756804 |
| troll | 0.035944045 |
| silly | 0.029536605 |
| hate | 0.013704538 |
| silly how do you study | 0.00325954 |
| spite | 0.002645433 |
| kid you have the absolute | 0.001619577 |

Table 16: Top ImageNet class-neutral concepts for "Crossword".

| "Crossword" – Top Concepts | Score |
| --- | --- |
| reading a newspaper | 0.30469692 |
| man reading a newspaper | 0.29045385 |
| crochet squares | 0.28316277 |
| sitting on a newspaper | 0.27749887 |
| newspaper sitting | 0.27106437 |
| crochet squares in a square | 0.2708223 |
| newspaper that has the words | 0.26934764 |
| holding a newspaper | 0.26375395 |
| newspaper while sitting | 0.26288068 |
| newspaper laying | 0.25538272 |
| square with a few crochet | 0.2461972 |
| square with a crochet | 0.24225119 |
| square of crochet squares | 0.23986068 |
| checkerboard | 0.23797607 |
| crochet blanket in a square | 0.2364017 |
| square of square crochet | 0.23590976 |
| on a newspaper | 0.2357213 |
| crochet square with a crochet | 0.23569846 |
| crochet blanket with a crochet | 0.23438567 |
| newspaper | 0.23315597 |
| crochet with a square | 0.23304509 |
| crochet square | 0.23259673 |
| newspaper sitting on | 0.23098715 |
| square of crochet yarn | 0.2291883 |
| square crochet | 0.22903368 |
| crochet blanket with a square | 0.22888878 |
| crochet square sitting | 0.22860557 |
| crochet in a square | 0.22856355 |
| square with a single crochet | 0.22752959 |
| free crochet | 0.22748157 |
| newspaper with | 0.22672665 |
| is on a newspaper | 0.2257084 |
| crochet blanket | 0.22547376 |
| checkered blanket | 0.22514643 |
| square of crochet | 0.22324148 |

Table 17: Top ImageNet class-neutral concepts for "Guacamole".

| "Guacamole" – Top Concepts | Score |
| --- | --- |
| tomatoes and avocado | 0.33948907 |
| avocado | 0.3125001 |
| nachos | 0.30761188 |
| ham and parsley | 0.2960047 |
| with avocado | 0.2952027 |
| colorful mexican | 0.28313732 |
| bacon and parsley | 0.27773544 |
| of avocado | 0.2754345 |
| side of salsa | 0.27470407 |
| peridot | 0.2734925 |
| salsa | 0.27124816 |
| nachos with | 0.27092364 |
| lime body | 0.27088284 |
| cheese and parsley | 0.26951405 |
| pile of limes | 0.26944226 |
| peas and bacon | 0.2671204 |
| limes and limes | 0.26682717 |
| lime cut | 0.2662533 |
| nachos with and | 0.26580203 |
| pasta with peas | 0.2626204 |
| tacos | 0.2619322 |
| pasta with ham and parsley | 0.26183394 |
| tomatoes and cilantro | 0.26161948 |

Table 18: Top ImageNet class-neutral concepts for "Bald Eagle".

| "Bald Eagle" – Top Concepts | Score |
| --- | --- |
| osprey flying | 0.2888234 |
| osprey | 0.27377927 |
| row of american flags | 0.26513425 |
| group of american flags flying | 0.25681192 |
| american flag and american flag | 0.23945728 |
| emu standing | 0.23752406 |
| yellowstone national | 0.23740456 |

Table 19: Top ImageNet class-neutral concepts for "Ballpoint Pen".

| "Ballpoint Pen" – Top Concepts | Score |
| --- | --- |
| wearing a pilot | 0.36252645 |
| markers | 0.36050144 |
| sharpie | 0.34563553 |
| stylus | 0.34399948 |
| pair of eyeglasses | 0.32426757 |
| notepad | 0.3228797 |
| calligraphy | 0.3222967 |
| paint and markers | 0.32198787 |
| close up of a needle | 0.32131955 |
| on a straw | 0.3209229 |
| and markers | 0.32090995 |
| eyeglasses | 0.31017447 |
| dots | 0.30764964 |
| crayons | 0.30598855 |

Table 20: Top ImageNet class-neutral concepts for "Coffeemaker".

| "Coffeemaker" – Top Concepts | Score |
| --- | --- |
| kettle sitting on | 0.43714887 |
| with a kettle | 0.4207235 |
| thermos | 0.40833473 |
| kettle | 0.40526068 |
| kettle sitting | 0.3881535 |
| stovetop maker | 0.37747166 |
| kettle kettle | 0.37671012 |
| kettle kettle kettle kettle | 0.37649006 |
| vases and vases | 0.37567452 |
| kettle kettle kettle | 0.37547356 |
| flask | 0.37485123 |
| kettle kettle kettle kettle kettle | 0.37305972 |
| large canister | 0.37186915 |
| flasks | 0.36940324 |
| decorative vases sitting | 0.3683877 |
| kitchen aid | 0.36793774 |
| milkshakes | 0.36605325 |
| large pottery | 0.36506 |
| set of kitchen | 0.3650242 |
| cookbook | 0.36469316 |
| vases sitting | 0.3643943 |

Table 21: Top ImageNet class-neutral concepts for "Doormat".

| "Doormat" – Top Concepts | Score |
| --- | --- |
| brick sidewalk | 0.37059835 |
| laying on gravel | 0.34978455 |
| laying on a carpeted | 0.34279323 |
| crochet blanket | 0.3421431 |
| sitting on a carpeted | 0.34104648 |
| laying on a step | 0.3400078 |
| crochet blanket in a square | 0.33642814 |
| brick walkway | 0.3319164 |
| carpeted floor | 0.33067068 |
| on a brick sidewalk | 0.32878387 |
| crochet blanket with a crochet | 0.32599914 |
| floor with a welcome | 0.32548892 |
| square of crochet yarn | 0.32428077 |
| crochet blanket made | 0.32336423 |
| carpeted staircase | 0.3227385 |
| dot blanket | 0.3213501 |
| mosaic floor | 0.32125634 |
| crocheted blanket | 0.31886423 |
| on a blanket | 0.3182252 |
| crochet blanket with a square | 0.31753486 |
| square of crochet squares | 0.3169018 |
| crochet squares | 0.3152317 |
| standing in a doorway | 0.3126963 |

Table 22: Top ImageNet class-neutral concepts for "Eraser".

| "Eraser" – Top Concepts | Score |
| --- | --- |
| crayons | 0.38733196 |
| chalk | 0.37349075 |
| graphite | 0.36778685 |
| crayon | 0.36178917 |
| charcoal | 0.351962 |
| lip balm lip | 0.35033816 |
| band aid cookie | 0.34825876 |
| lip balm | 0.3409983 |
| nose sticking | 0.34016216 |
| sticking | 0.33971623 |
| markers | 0.33940658 |
| matchbox | 0.33480892 |
| wand | 0.33415005 |
| stylus | 0.3318595 |
| band aid card | 0.33174193 |
| toothbrush | 0.33009088 |
| office supplies | 0.32956824 |
| band aid flexible | 0.32946587 |

Table 23: Top ImageNet class-neutral concepts for "American Lobster".

| "American Lobster" – Top Concepts | Score |
| --- | --- |
| pasta with shrimp | 0.3148532 |
| adirondack sitting | 0.31178916 |
| large shrimp | 0.3085378 |
| shrimp and pasta | 0.30821544 |
| shrimp cooking | 0.3032636 |
| large cast cooking | 0.30282205 |
| pasta with shrimp and cheese | 0.2984723 |
| adirondack | 0.29169592 |
| and mussels | 0.2904386 |
| legs and other seafood | 0.28870153 |
| close up of a shrimp | 0.2871693 |
| of pasta with shrimp | 0.2823377 |
| seafood | 0.2808096 |
| mussels | 0.28018713 |
| cast cooking | 0.27951774 |
| shrimp and cheese | 0.27841187 |
| shrimp | 0.2726224 |

Table 24: Top ImageNet class-neutral concepts for "Fire Truck".

| "Fire Truck" – Top Concepts | Score |
|---|---|
| firefighter spraying | 0.34155905 |
| group of firefighters | 0.3347583 |
| firefighters | 0.33051446 |
| firefighter | 0.31450543 |
| firefighter wearing | 0.2843979 |
| hydrant spraying | 0.26793447 |
| firefighter wearing a | 0.26590723 |
| firefighter cuts | 0.2613345 |
| farmall parked | 0.2591076 |
| dashboard with flames | 0.25825307 |
| flames painted | 0.2540929 |

