# OpenReview forum: "WASP: A Weight-Space Approach to Detecting Learned Spuriousness"
_NeurIPS.cc/2025/Workshop/Reliable_ML — NeurIPS 2025 - Reliable ML Workshop_

### Official Review · Reviewer_V4je · 2025-09-19
**A novel method which properly address previous works' limitations**

**Rating:** 6
**Confidence:** 3

**Review:**

# Summary
The author provides a weight-space bias detection method for foundation models that can be applied to various modalities and used to detect spurious features beyond training and validation sets.

* quality: fair
* clarity: good
* originality: good
* significance: fair

# Strengths
- A novel weight-centric approach that overcomes existing methods' shortcomings
  - As noted by authors, most of the previous approaches for bias-discovery centered on data, and the model's representation/output of data.
  - These approaches usually can not detect spurious features that are beyond the coverage of the development phase seen dataset.
  - The proposed method exactly tackles this problem by focusing on the training dynamics of the model weight.
- The proposed method WASP indeed shows improved performance in the worst-case group compared with the baseline across multiple considered datasets.
- The authors provide insightful qualitative results on detected spurious correlation.

# Weaknesses
- As the method involved training, it may require more computational resources compared to training-free baseline methods, but the authors do not include them.
- The method works by training the final head layer on top of frozen representations. However, there are many recent works showing that the final layer representation is not so informative for the current mainstream models, i.e., autoregressive LLMs as well as multimodal LLMs, rather, intermediate representations are much more important for downstream tasks (Skean et al. 2025).
  - It is unclear whether the proposed method can be applicable to the autoregressive LLMs or MLLMs as well.


# Suggestions for Authors
- Please address the weaknesses.

---

> Reference
* Layer by Layer: Uncovering Hidden Representations in Language Models, Skean et al. 2025

---

### Official Review · Reviewer_2Kwc · 2025-09-20
**An effective algorithm for detecting spurious correlations**

**Rating:** 6
**Confidence:** 2

**Review:**

### Summary:

The paper introduces an algorithm they call WASP, which detects a model's learnt correlations after fine-tuning. This algorithm looks at the drift of weights during fine-tuning and detects whether a spurious correlation has been formed between chosen concepts. They test this algorithm on a range of datasets and a number of models.

### Strengths:

The introduced algorithm seems to work well in experimentation. An overall interesting contribution worth presenting at this workshop.

### Weaknesses:

I don't understand some details of the algorithm, though I am not familiar with the spurious correlation literature (the extensive related work section in this paper was much appreciated).

In a linear optimization problem, the initialization typically doesn't change the solution. I'm assuming that it does here because the optimization problem is so underspecified. With a large enough dataset equal to the size of the embedding space, initializing the weights to M(c_i) certainly wouldn't change the ERM solution at all, making part of step 1 useless. It would be helpful for the authors to elaborate on what they mean by run ERM in step 1.

It would also be helpful if there was a justification of why, after training, you would expect that w_k^* will have anything to do M(c_i). It's not obvious to me why the weights of the last linear layer would semantically be comparable to the embedding of a class, even though they lie in the same space. As someone who doesn't do work in this area, this section confuses me, and I would have appreciated some more explanation of these details in section 3.

### Suggestions for authors:

I was initially unsure of what problem the authors were attempting to solve, it would be helpful to have lines 64-69 appear in the introduction instead.